# Clinical Application of Comprehensive Genomic Profiling Tests for Diffuse Gliomas

**DOI:** 10.3390/cancers14102454

**Published:** 2022-05-16

**Authors:** Takaki Omura, Masamichi Takahashi, Makoto Ohno, Yasuji Miyakita, Shunsuke Yanagisawa, Yukie Tamura, Miyu Kikuchi, Daisuke Kawauchi, Tomoyuki Nakano, Tomohiro Hosoya, Hiroshi Igaki, Kaishi Satomi, Akihiko Yoshida, Kuniko Sunami, Makoto Hirata, Tatsunori Shimoi, Kazuki Sudo, Hitomi S. Okuma, Kan Yonemori, Hiromichi Suzuki, Koichi Ichimura, Yoshitaka Narita

**Affiliations:** 1Department of Neurosurgery and Neuro-Oncology, National Cancer Center Hospital, Tokyo 1040045, Japan; tomura@ncc.go.jp (T.O.); masataka@ncc.go.jp (M.T.); mohno@ncc.go.jp (M.O.); ymiyakit@ncc.go.jp (Y.M.); shuyanag@ncc.go.jp (S.Y.); yuktamur@ncc.go.jp (Y.T.); miykikuc@ncc.go.jp (M.K.); dkawauch@ncc.go.jp (D.K.); tomnakan@ncc.go.jp (T.N.); thosoya@ncc.go.jp (T.H.); 2Department of Radiation Oncology, National Cancer Center Hospital, Tokyo 1040045, Japan; hirigaki@ncc.go.jp; 3Department of Pathology, Kyorin University School of Medicine, Tokyo 1818611, Japan; kaishi-satomi@ks.kyorin-u.ac.jp; 4Department of Diagnostic Pathology, National Cancer Center Hospital, Tokyo 1040045, Japan; akyoshid@ncc.go.jp; 5Department of Laboratory Medicine, National Cancer Center Hospital, Tokyo 1040045, Japan; ksunami@ncc.go.jp; 6Department of Genetic Medicine and Services, National Cancer Center Hospital, Tokyo 1040045, Japan; mahirata@ncc.go.jp; 7Department of Medical Oncology, National Cancer Center Hospital, Tokyo 1040045, Japan; tshimoi@ncc.go.jp (T.S.); ksudo@ncc.go.jp (K.S.); hsumiyos@ncc.go.jp (H.S.O.); kyonemor@ncc.go.jp (K.Y.); 8Division of Brain Tumor Translational Research, National Cancer Center Research Institute, Tokyo 1040045, Japan; hiromics@ncc.go.jp; 9Department of Brain Disease Translational Research, Faculty of Medicine, Juntendo University, Tokyo 1138421, Japan; k.ichimura.uk@juntendo.ac.jp

**Keywords:** glioma, genomic profiling test, clinical actionability, germline mutations

## Abstract

**Simple Summary:**

Cancer patients suffer from recurrence after the completion of standard treatments and exhaustion of treatment options. The comprehensive genomic profiling test (CGPT) is a platform that enables those patients to access the eligible promising therapeutic agents based on their genomic aberrations, using next-generation sequencing. Though CGPTs have been utilized since 2019 in Japan, only limited findings have been available about their use for glioma patients. The aim of this study was to reveal the comprehensive results of CGPT in glioma patients, especially the clinical actionability, which means the probability of being able to receive appropriate molecular targeting therapeutic agents. In our cohort, the clinical actionability was 18.5%, which was compatible with the results of previous reports for tumors other than glioma. We confirmed that CGPT is also useful for glioma patients, and our result will encourage a future increase of CGPT use in our clinical practices.

**Abstract:**

Next-generation sequencing-based comprehensive genomic profiling test (CGPT) enables clinicians and patients to access promising molecularly targeted therapeutic agents. Approximately 10% of patients who undergo CGPT receive an appropriate agent. However, its coverage of glioma patients is seldom reported. The aim of this study was to reveal the comprehensive results of CGPT in glioma patients in our institution, especially the clinical actionability. We analyzed the genomic aberrations, tumor mutation burden scores, and microsatellite instability status. The Molecular Tumor Board (MTB) individually recommended a therapeutic agent and suggested further confirmation of germline mutations after considering the results. The results of 65/104 patients with glioma who underwent CGPTs were reviewed by MTB. Among them, 12 (18.5%) could access at least one therapeutic agent, and 5 (7.7%) were suspected of germline mutations. A total of 49 patients with *IDH*-wildtype glioblastoma showed frequent genomic aberrations in the following genes: *TERT* promoter (67%), *CDKN2A* (57%), *CDKN2B* (51%), *MTAP* (41%), *TP53* (35%), *EGFR* (31%), *PTEN* (31%), *NF1* (18%), *BRAF* (12%), *PDGFRA* (12%), *CDK4* (10%), and *PIK3CA* (10%). Since glioma patients currently have very limited standard treatment options and a high recurrence rate, CGPT might be a facilitative tool for glioma patients in terms of clinical actionability and diagnostic value.

## 1. Introduction

Glioma is a common malignant brain tumor that arises within the central nervous system. Among gliomas, glioblastoma (GBM) is a malignant neoplasm with a 5-year survival rate of 10–15% [1,2]. One of the reasons for this low survival rate is because of the lack of promising therapeutic drugs except for temozolomide (TMZ) and bevacizumab. Though some advances have been made for the limited entity of gliomas [3], no effective new drugs have been developed against malignant gliomas for over 10 years. This is problematic because of the absence of any second-line drugs. Difficulties in producing therapeutic drugs based on the genomic profile of individual glioma patients are the main reason for the scarcity [4].

In Japan, the Center for Cancer Genomics and Advanced Therapeutics was established in 2018 to promote the management and utilization of cancer information collected from selected medical institutions and hospitals that offer therapy based on individual genomic profiles [5]. The comprehensive genomic profiling test (CGPT) has been used since June 2019 under public health insurance coverage, enabling us to find suitable drugs for individual patients. Currently, three CGPTs using next-generation sequencing (NGS) panels are available within the public medical insurance coverage in Japan: OncoGuide^TM^ NCC OncoPanel System (NOP) (National Cancer Center, Tokyo, Japan, and Sysmex Corporation, Kobe, Japan) FoundationOne^®^ CDx Cancer Genomic Profile (F-One) (Cambridge, MA, USA), and FoundationOne^®^ Liquid CDx Cancer Genomic Profile (F-One liquid) (Cambridge, MA, USA) (Appendix A).

The National Cancer Center and Sysmex Corporation developed the NOP to detect mutations in 124 genes, 12 fusion genes, and tumor mutation burden (TMB) scores. The NOP compares tumor and blood DNA; therefore, it can detect mutations from a small sample obtained using a needle biopsy. F-One is distributed from Roche (Basel, Switzerland) and can detect mutations in 324 genes, 36 fusion genes, microsatellite instability (MSI), and TMB scores. NOP and F-One are based on the NGS of tumor DNA; however, F-One liquid analyzes only circulating tumor DNA.

The feasibility of NOP was validated by the Trial of Onco-Panel for Gene-profiling to Estimate both Adverse Events and Response during cancer treatment (TOP-GEAR project) [6], before its clinical application. In the TOP-GEAR project, 13.3% of the enrolled patients received therapeutic agents, including 8% with investigational drugs, 3% with approved drugs, and 2% with off-label drugs [6]. Another representative result of CGPTs to assess clinical actionability was demonstrated in 10,000 MSK-IMPACT study patients [7]. In this study, 10.5% of patients were able to receive the therapeutic agent.

Given the results of similar studies (including the abovementioned ones), the probability of being successfully assigned to any therapeutic agent is approximately 10–20% in both Japan and the United States. As public health insurance starts covering cancer CGPT, its usage might increase the number of new molecular targeted agents for brain tumors, especially GBMs and recurrent malignant gliomas. In this study, we report the analysis of 104 glioma cases, including 49 GBMs, using the NGS panel and discuss the contemporary significance and prospects of CGPTs in glioma. To our knowledge, this is the first report on CGPT results of glioma patients in Japan, and our purpose is to reveal the real practical clinical actionability.

## 2. Materials and Methods

### 2.1. Patients

All glioma patients treated at our institution were candidates for CGPTs. The patients meeting the following three criteria were suitable to undergo CGPTs:(1)Termination of standard treatment because of local progression, including those expecting termination.(2)Eligibility for alternate drug therapy after CGPT was confirmed by an attending physician after an examination.(3)A Karnofsky Performance Status ≥70. This criterion implicates that they can attend outpatient clinics.

Written informed consent for the use of genomic and clinical data for research purposes was obtained from all participants. The tests were performed under either regular medical insurance or clinical trials (A prospective clinical registry study of genetic profiling and targeted therapies in patients with rare cancers; MASTER KEY Protocol). The clinical trial was approved by the NCC Institutional Review Board (No. 2016-460 [NCCH1612]).

### 2.2. Tissue Diagnosis

Two pathologists histologically confirmed the diagnosis based on the 2016 World Health Organization (WHO) Classification of Tumors of the Central Nervous System. Regarding the genetic aberrations (WHO 2016 classification), each diagnosis was reclassified into the latest classification based on the 2021 WHO Classification of Tumors of the Central Nervous System (WHO 2021 classification) [8]. The 1p19q co-deletion, essential for diagnosing oligodendroglioma, was assessed by either fluorescence in situ hybridization or multiplex ligation-dependent probe amplification [9].

### 2.3. Preparation of Tumor Tissue for CGPT

F-One and NOP were used to detect genomic aberrations in DNA extracted from a small tissue sample. A total of 5 slides of 10-μm sections or 10 slides of 4-μm to 5-μm sections were prepared for both from formalin-fixed paraffin-embedded tumor tissues and submitted for DNA extraction. In NOP, a blood sample was also submitted as a paired sample. A pathologist at our institution determined the tumor purity of each section, and those with higher tumor purity than the manufacturers’ recommendations were analyzed: tumor purity ≥30% for F-One, and ≥20% for NOP.

### 2.4. Analysis of the CGPT Results

Both F-One and NOP utilized NGS to obtain genomic aberrations (mutation, copy number alteration, rearrangements, fusion, and other structural variants) and MSI and TMB scores. TMB was defined as −1 for cases with no TMB score in the manufacturers’ reports. As the NGS raw data was unavailable in F-One and NOP, the report was used to analyze the number of genes with genomic aberrations. The variants of unknown significance reported in F-One were excluded in the subsequent analysis as their biological significance was unclear, and they were not actionable genomic aberrations. Since NOP used blood samples, germline genomic aberrations were also detected and compared using the tumor sample data. If a gene harbored multiple mutations or other alterations, they were integrated and counted as one; for example, if *TP53* S240R and *TP53* V73fs*76 were detected in the same sample, then the number of genomic aberrations on *TP53* was counted as one. ComplexHeatmap was used for visualizing data as oncoprints [10].

### 2.5. Molecular Tumor Board, Clinical Actionability, and Germline Mutations

The Molecular Tumor Board determined the therapeutic agents and their recommended levels from the CGPTs based on the oncological significance of the genomic aberrations. The attending physician and patient discussed the proposed agent and made the final decision. NOP, which compares peripheral blood DNA with tumor tissue DNA, detects germline mutations. However, in F-One, the Molecular Tumor Board selected presumed germline mutations according to the operation guideline in the proposal concerning the information transmission process in genomic medicine (AMED, Japan; see https://www.amed.go.jp/content/000064662.pdf for details), and proposed remarks regarding the need for genomic counseling or further testing.

## 3. Results

### 3.1. Demographics of the Enrolled Patients

In the study, we enrolled 104 diffuse glioma patients who underwent CGPT (F-One or NOP) at our institution from June 2019 to January 2022 (Table 1). A total of 39 and 65 patients underwent CGPTs as a clinical study and within the public health insurance coverage, respectively. Of the patients, 94 and 10 patients used F-One and NOP, respectively. Each histological classification was based on WHO 2021 classification. All recurrent patients received radiotherapy and temozolomide. The number of cases (number of recurrent cases is described in the parentheses) was as follows: 19 isocitrate dehydrogenase (*IDH*)-mutant (mt) astrocytoma 11 (3) cases of grade 3 and 8 (5) cases of grade 4; 49 (12) *IDH*-wild type (wt) GBM; 6 (0) *IDH*-wt diffuse astrocytoma; 6 (0) diffuse midline glioma, H3 K27-altered; 23 oligodendroglioma, 6 (0) cases of grade 2 and 17 (6) cases of grade 3; 1 (0) pilocytic astrocytoma. (For detailed data, please refer to the Appendix A). 

### 3.2. Genomic Aberrations, TMB, and MSI

Genomic aberrations and TMB scores are displayed in Figure 1. TMB scores were available in only 65 patients who underwent CGPTs within the public health insurance coverage. Although F-One has a companion diagnostic function for pembrolizumab [11], no patient was diagnosed with high MSI (MSI-H). However, there were three cases with remarkably high TMB scores (>40 mutations mt/Mb).

A patient was diagnosed with anaplastic oligodendroglioma with metastases to the cranial and femoral bones and had a TMB score of 72 mt/Mb. Another patient has recurrent anaplastic oligodendroglioma with a TMB score of 44 mt/Mb, and the Molecular Tumor Board suggested the possibility of TMZ-induced hypermutation. There was also a patient with newly diagnosed GBM, a TMB score of 101 mt/Mb, and mutations in *MSH6* (*MSH6* C694fs*4 and *MSH6* I795fs*15). The Molecular Tumor Board suggested that either of the *MSH6* mutations might be originated from a germline variant, and the patient underwent genetic counseling.

### 3.3. Frequency of Gene Aberrations and Histological Classification

The median number of gene aberrations in the 104 cases of diffuse glioma was 5 (interquartile range [IQR]: 4–7). The median number of genomic aberrations in newly diagnosed and recurrent *IDH*-mt astrocytoma was 4 (IQR: 3.5–6) and 5.5 (IQR: 3.5–8.25), respectively, and that of oligodendroglioma was 5 (IQR: 4–8) and 5 (IQR: 3.5–5.75), respectively. The number of genomic aberrations in newly diagnosed and recurrent *IDH*-wt GBM was 5 (IQR: 4–8) and 5.5 (IQR: 3.75–6), respectively. Most of the recurrent cases were treated with TMZ. However, the number of newly diagnosed and recurrent cases was not significantly different (*p*-value = 0.43, Mann-Whitney U test).

Among the 49 *IDH*-wt glioblastomas, *TERT* promoter (67%), *CDKN2A* (57%), *CDKN2B* (51%), *MTAP* (41%), and *TP53* (35%) were detected (Figure 2A). In the 19 *IDH*-mt astrocytoma cases, *TP53* (95%), *ATRX* (63%), *MLL2* (21%), and *CDKN2A/B* homozygous deletion (16%) were detected (Figure 2B). In the 23 oligodendroglioma cases, *CIC* (70%), *PIK3CA* (35%), *PIK3R1* (26%), and *FUBP1* (22%) were detected (Figure 2C). In the 49 GBM patients, we compared the relationship between gene aberrations with and without recurrence (Figure 3). The representative top five genes reported in the newly diagnosed 37 GBM cases, including the primary surgery specimens, were *TERT* (65%), *CDKN2A* (57%), *CDKN2B* (51%), *TP53*(32%), and *MTAP* (43%). In the 12 recurrent GBM cases, the top five genes reported were *TERT* (75%), *EGFR* (42%), *TP53* (42%), *NF1* (50%), *CDKN2A* (58%), and *CDKN2B* (50%).

### 3.4. Histological Re-Classification Based on CGPTs

Among GBMs, two cases were originally diagnosed as anaplastic astrocytoma according to WHO 2016 classification; however, CGPT revealed that they harbored *EGFR* amplification and *TERT* promoter mutation, and they were reclassified as GBM based on WHO 2021 classification. None of the other six *IDH*-wt astrocytoma cases were reclassified into new entities because none of them possessed the *MYC* or *PDGFR* amplification. None of them was classified as diffuse pediatric-type high-grade glioma and H3-wt.

### 3.5. Clinical Actionability

The clinical actionability in the 31 *IDH*-wt GBMs reviewed by the Molecular Tumor Board is shown in Table 2. Of these 31 patients, 3 (9.7%) were enrolled in clinical trials and had access to therapeutic drugs. Another four (12.9%) patients could potentially have therapeutic options. This implies that they had targetable genomic aberrations, but other preferable options (e.g., bevacizumab, enrollment in other more promising clinical trials) should be prioritized before enrolling in the clinical trials. Therefore, seven patients (22.6%) were proposed available clinical trials in which they could use specific molecular targeted drugs based on their genomic aberrations. The candidate therapeutic agents were dabrafenib/trametinib (*n* = 3), *FGFR* inhibitors (*n* = 3), and pembrolizumab (*n* = 1) (Table 3).

For oligodendroglioma, ATR inhibitors, *IDH1* inhibitors, and pembrolizumab were recommended (*n* = 1 for each). An *IDH1* inhibitor for Grade 3 (*n* = 1) *IDH*-mt astrocytoma and *FGFR* inhibitors and pembrolizumab for diffuse midline glioma (*n* = 1) were recommended (Appendix A).

Dabrafenib/trametinib was available through the “Managed Access Program” provided by Novartis, and all three patients (58_GBM, 63_GBM, 65_GBM) used dabrafenib/trametinib under enrollment in the clinical trial (The prospective trial of patient-proposed healthcare services with multiple targeted agent based on the result of gene profiling by multigene panel test (BELIEVE), NCC Certified Clinical Research Review Board approval; NCCH1901) [12].

### 3.6. Germline Mutations and Subsequent Genetic Counseling

Five patients (7.7%) potentially had germline mutations (Table 3). They comprised one patient with *NF1*, two with *MSH6*, and two with *BRCA2* mutations. All the patients were highly recommended to undergo genetic counseling in our institution.

## 4. Discussion

We conducted a comprehensive analysis of glioma patients using two CGPTs. Our findings revealed that 7.7% of the patients were suspected to have germline mutations.

The most critical result of CGPT is the number of therapeutic agents available and not the genomic aberrations. Compared with patients with *IDH*-mt astrocytoma or oligodendroglioma, those with *IDH*-wt GBM have more difficulty finding therapeutic agents. This is because of the relatively larger number of ongoing clinical trials targeting the mutant *IDH* [13,14,15]. According to the previous studies in which tumors other than gliomas were included, 10.5–17.8% of the patients might receive therapeutic agents [6,7,16]. In our cohort, three *IDH*-wt GBM patients (9.7%) could be enrolled in ≥1 clinical trial to use molecularly targeted agents (Table 2). Few clinical trials, other than those of *IDH*-inhibitors, are available for patients with *IDH*-mt gliomas after completion of the standard treatments; however, detailed genomic analysis has been reported previously [17]. In our cohort, among 34 patients with *IDH*-mt gliomas, only two (5.8%) were suggested therapeutic agents other than *IDH*-inhibitors (Appendix A). Clinical trials with promising molecularly targeting agents other than *IDH*-inhibitors are needed for GBM and IDH-mt gliomas.

CGPT is a helpful clinical diagnostic tool to determine the integrated pathological classification based on the WHO 2021 classifications of central nervous system tumors. In the 2021 WHO classification, the mutational status of the established diagnostic genes must be inspected molecularly, including *IDH1/2*, *CDKN2A/B*, *H3F3A*, *EGFR*, and chromosomal 1p19q co-deletion. Currently, each mutational status is obtained at an individual institutional laboratory. However, in some institutions, molecular diagnosis cannot be performed because of the lack of equipment and facilities. CGPTs can perform and compensate for the screening of these established genomic aberrations, which should be obtained for clinical diagnosis. In our cohort, among the eight cases with *IDH*-wt astrocytoma, two were molecularly reclassified into GBM based on CGPT results, though they were originally categorized as anaplastic astrocytoma and Grade 3 *IDH*-wt in the WHO 2016 classification of CNS tumor. However, the remaining six cases were not assigned any classification, and they were defined as not elsewhere classified (NEC). For these cases, methylation status analysis is a possible tool for further categorization [18]. Given that it usually takes 3 to 4 weeks to receive the review from the Molecular Tumor Board after submitting CGPTs, the use of CGPTs should be considered in the early clinical course of patients with glioblastoma whose prognosis is only 15 months.

There is a discrepancy between the demographic composition of our cohort and those of previous studies. We analyzed selected glioma patients who satisfied the criteria for CGPTs. The frequency of each genomic aberration of our study was not significantly different from previously reported representative results of The Cancer Genome Atlas glioma cohort [19,20,21] (Figure 4), except for the relatively higher frequency of genetic aberrations in *BRAF* and *FGFR*s. The frequency of genomic aberrations for *FGFR1* (4%), *FGFR2* (0%), *FGFR3* (8%), and *FGFR4* (4%) is lower than other MAPK-activating genes such as *EGFR* (31%) and *PTEN* (31%). However, the total proportion of mutations in the FGFR family is not negligible (16%). Therefore, pan-FGFR inhibitors might effectively treat GBMs [22,23,24]. The frequency of *BRAF* mutation in our study is higher than that reported in previous reports because patients must have CGPT to attend a clinical trial using *BRAF* inhibitors after the detection of *BRAF* V600E using immunohistochemistry.

The patients’ access to therapeutic agents is an important factor to be considered. All patients who could receive any therapeutic agent were enrolled in this clinical study. The patients could not use those agents under general healthcare services. Therefore, patient-proposed healthcare services are a unique system for off-label medications in Japan designed to efficiently avoid loss of opportunities because of the time lag between clinical trial closing and drug approval by Pharmaceutical and Medical Device Agencies. This system is requisite for rare cancers, including gliomas, because orphan drug clinical trials are not ubiquitously accessible. Therefore, the National Cancer Center in Japan offers a basket platform of clinical trials for these patient-proposed healthcare services [25,26].

Pembrolizumab was a possible therapeutic agent for three patients with high TMB scores (100_DMG, 101_GBM, and 119_OLG). MSI-H is a sufficient condition for administering pembrolizumab, and the existence of such targeted molecular aberration is mandatory for applying those molecularly targeting agents. However, no patient was diagnosed with MSI-H (among 65 patients) in this study, which is compatible with previous literature [27,28,29]. Although a daily clinical practice, the TMB score is not a definite prognostic indicator to assess the immune checkpoint inhibitors (e.g., pembrolizumab) applicable in glioma patients; the temozolomide-induced hypermutation increases the TMB scores in glioma [30,31].

Clinicians should carefully provide accurate information on germline mutations and secondary results of CGPTs. As shown in Table 3, five patients presumably harbored germline mutations, including four *IDH*-wt glioma patients. Patients with suspected germline mutations generally consult genetic counseling with their families. However, not all patients get a definitive diagnosis because of their apprehension regarding the influence of the results on their families. We therefore should explain to patients beforehand that CGPT might unveil hidden germline mutations.

Some limitations to this study are to be recognized. First, the sample of this study was small because of the selection of participants being limited to those with a good performance status. Our result may only reflect the results of the very limited patients who sufficed the criteria, and thus cannot be applied to all the other glioma patients. However, the patient inclusion criteria used in this paper are the ones that need to be met when glioma patients consider enrolling in clinical trials. Therefore, our result could be one of the representative findings regarding the clinical actionability in glioma patients who are potentially eligible for clinical trials at this moment. Second, NOP and F-One did not uncover all mutations and detect fusion genes such as FGFR or MET fusions. Genes of interest and algorithms of mutation call differ between F-One and NOP. These limitations may lead to underestimating the composite of genomic aberrations in fragile patients. Increasing the number of CGPT uses, especially in the early clinical course of the glioma patients, would overcome these limitations in the future.

## 5. Conclusions

CGPTs are helpful to detect driver genomic aberrations; this enables glioma patients to participate in clinical trials and the categorization of these patients based on the WHO 2021 classification. For patients with glioma, which currently has very limited standard treatment options and a high recurrence rate, CGPTs should be considered in their early clinical courses. Considering and improving the clinical actionability, we hope more clinically actionable glioma targeting CGPT will be developed in the future.

## Figures and Tables

**Figure 1 cancers-14-02454-f001:**
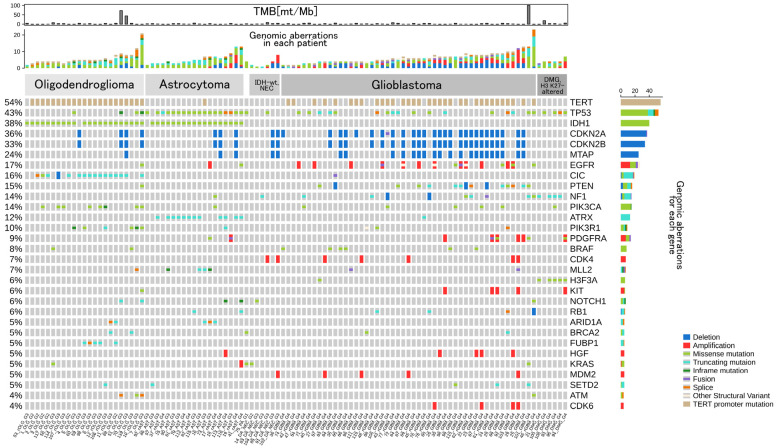
Oncoprint illustrating genomic aberrations observed in 104 patients with glioma using a comprehensive genomic profiling test (CGPT). The top 30 most frequently reported genes are displayed, and their type of genomic aberrations are categorized as follows by their colors: Deletion, Amplification, Missense mutation, Truncating mutation, Inframe mutation, Fusion, Splice, Other Structural Variant, and *TERT* promoter mutation. The number of genomic aberrations for each case and gene is displayed at the middle row and the right of the oncoprint, respectively. At the top, the tumor mutation burden (TMB, [mutations/Mb]) score for each patient is shown. ID for each patient is shown at the bottom. For the cases with an unavailable TMB score, TMB is shown as −1. See also Appendix A. for a full genomic display.

**Figure 2 cancers-14-02454-f002:**
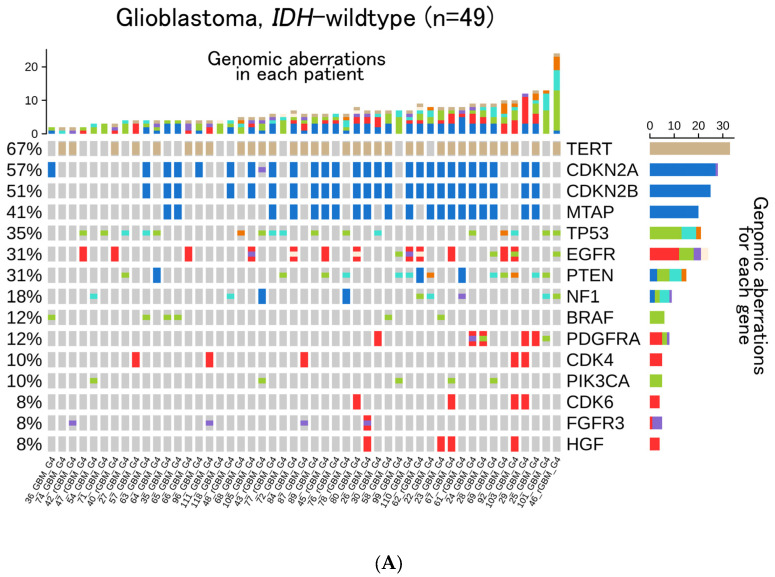
Oncoprint of (**A**) 49 patients with Glioblastoma, *IDH*-wildtype. The top 15 most frequently reported genes: (**B**) 19 patients with Astrocytoma, *IDH*-mutant, Grade 2 or 3. The top 15 most frequently reported genes; (**C**) 23 patients with Oligodendroglioma, *IDH*-mutant, and 1p/19q-co-deleted, Grade 2 or 3. The top 15 most frequently reported genes are displayed respectively.

**Figure 3 cancers-14-02454-f003:**
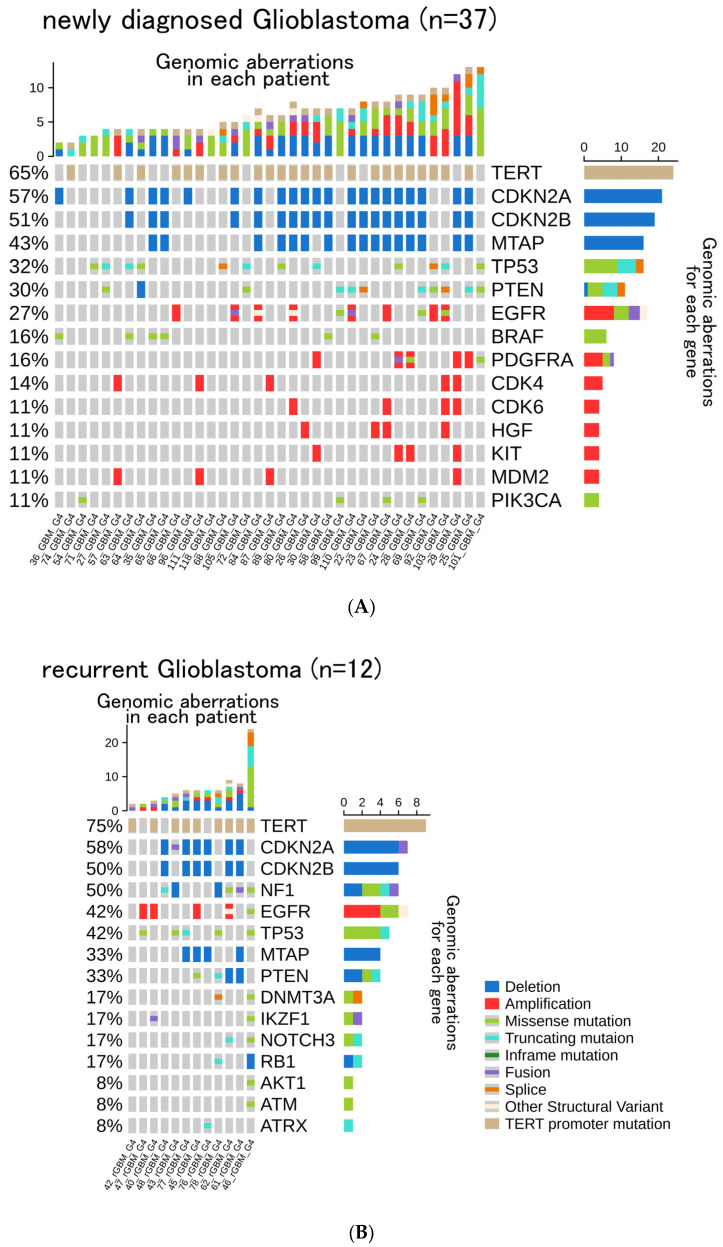
Oncoprint of (**A**) 37 patients with newly diagnosed *IDH*-wt GBM; and (**B**) 12 patients with recurrent *IDH*-wt GBM. The top 15 most frequently reported genes are displayed, respectively.

**Figure 4 cancers-14-02454-f004:**
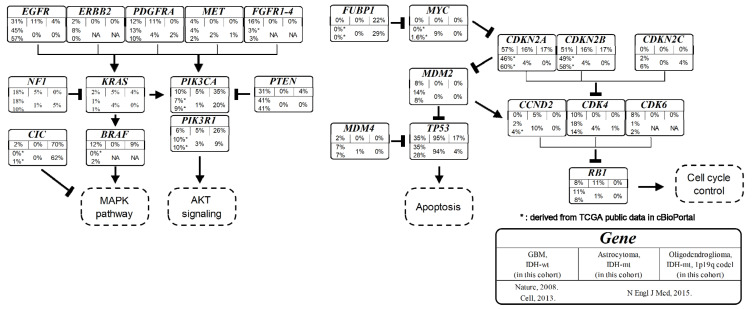
Results of frequently reported genes in genomic profiling test or next-generation sequencing in glioma. Frequently reported genes are categorized into representative signaling pathways, and their frequency is illustrated in each box entitled to the gene. In each box, (I) the top row indicates the name of the gene; (II) the middle row indicates the frequency of genomic aberrations in our cohort for the selected histological classification groups, such as *IDH*-wt glioblastoma, *IDH*-mt astrocytoma, and oligodendroglioma *IDH*-mt 1p19q co-deleted; (III) the bottom row indicates the reference study. (*): missing data in the original reference, and the corresponding frequency is derived manually from cBioPortal (https://www.cbioportal.org/, accessed on 1 February 2022). *Nature*, 2008 (reference [20]): *Cell*, 2013 (reference [21]), *N. Engl. J. Med.*, 2015 (reference [19]).

**Table 1 cancers-14-02454-t001:** Demographics of the enrolled patients with glioma who underwent genomic profiling test (CGPT). (*n* = 104).

Characteristic	Subset	*n* Total (Recurrent)	Median Age (Range) [y]
Gender	male	63 (15)	49 (24–91)
	female	41 (11)	59 (22–72)
Genomic Profiling Test	FoundationOne CDx Cancer Genomic Profile	94 (24)	49.5 (24–91)
	OncoGuide NCC OncoPanel System	10 (2)	49 (22–72)
Diagnosis (WHO 2021)	Glioblastoma, *IDH*-wildtype	49 (12)	56 (27–91)
	Diffuse Astrocytoma, *IDH*-wildtype, NEC	6 (0)	49 (37–64)
	Astrocytoma, *IDH*-mutant		
	Grade 4	8 (5)	40.5 (31–52)
	Grade 3	11 (3)	46 (29–74)
	Oligodendroglioma, *IDH*-mutant, and 1p/19q-codeleted		
	Grade 3	17 (6)	46 (29–74)
	Grade 2	6 (0)	49 (35–72)
	Diffuse midline glioma, H3 K27-altered	6 (0)	36.5 (30–45)
	Pilocytic astrocytoma	1 (0)	22 (NA)

Histological classification was re-defined based on the 2021 WHO Classification of Tumors of the Central Nervous System (WHO 2021). The text in parentheses describes the number of recurrent cases for each histological classification. IDH: isocitrate dehydrogenase; NEC: not elsewhere classified.

**Table 2 cancers-14-02454-t002:** Clinical actionability in the 31 *IDH*-wt GBMs of which the Molecular Tumor Board reviewed the results of CGPT.

No.	ID (in this Cohort)	CGPT	Diagnosis	Age	Gender	Acctionable Gene Aberration	Therapeutic Agents	Drug Type
1	58_GBM	F-One	GBM, *IDH*-wt	40	M	*BRAF* V600E	Dabrafenib/Trametinib	clinical trial
2	63_GBM	F-One	GBM, *IDH*-wt	62	F	*BRAF* V600E	Dabrafenib/Trametinib	clinical trial
3	65_GBM	F-One	GBM, *IDH*-wt	52	F	*BRAF* V600E	Dabrafenib/Trametinib	clinical trial
4	71_GBM	F-One	GBM, *IDH*-wt	55	F	*FGFR1* K656E	*FGFR* inhibitor	Investigational drug *
5	87_GBM	F-One	GBM, *IDH*-wt	59	M	*FGFR3 FGFR3-TACC3* fusion	*FGFR* inhibitor	Investigational drug *
6	101_GBM	F-One	GBM, *IDH*-wt	59	F	*MSH6* C694fs*4, *MSH6* I795fs*15	Pembrolizumab	clinical trial *
7	111_GBM	F-One	GBM, *IDH*-wt	53	M	*FGFR3 FGFR3-TACC3* fusion	*FGFR* inhibitor	Investigational drug *

Seven patients could be assigned to at least one clinical trial. Off-label use: use of the therapeutic agent within the coverage of public health insurance that is already approved for other tumors; however, not yet approved for glioma. GBM: glioblastoma, CGPT: comprehensive genomic profiling test, F-One: FoundationOne^®^ CDx Cancer Genomic Profile, wt: wildtype, (*): enrollment in a clinical trial is pending, considering other therapeutic options.

**Table 3 cancers-14-02454-t003:** Presumed germline mutations based on the CGPT results.

No.	ID (in this Cohort)	CGPT	Diagnosis	Age	Gender	Gene Aberration	Blood Sample Diagnosis	Therapeutic Drug Accesibility
1	92_GBM	F-One	GBM, *IDH*-wt	91	M	*BRCA2* R2318*	no	no
2	101_GBM	F-One	GBM, *IDH*-wt	59	F	*MSH6* C694fs*4, *MSH6* I795fs*15	no	Yes (Pembrolizumab) *
3	104_DA_NEC	NOP	Diffuse Astrocytoma, *IDH*-wt, NEC	51	F	*NF1* Q2434*	Yes	no
4	115_AST	F-One	Astrocytoma, *IDH*-mt	51	M	*MSH6* Y524fs*46	no	no
5	119_OLG	F-One	Oligodendroglioma, *IDH*-mt	52	F	*BRCA2* I682fs*48	no	Yes (Pembrolizumab/Olaparib) *

NEC: Not Elsewhere Classified, NOP: OncoGuide^TM^, NCC OncoPanel System, NOP can confirm germline mutations by comparing peripheral blood DNA and tumor tissue DNA. (*): enrollment in a clinical trial is pending, considering other therapeutic options.

## Data Availability

The data presented in this study are available on request from the corresponding author. The data are not publicly available due to privacy restrictions.

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
