# Peer review of "Clinical Application of Comprehensive Genomic Profiling Tests for Diffuse Gliomas"

_cancers, 2022, doi:10.3390/cancers14102454_

Round 1
Reviewer 1 Report
The authors present an interesting manuscript. Some aspects should be considered before accepting the manuscript for publication:
-Line 51. I agree that the low survival rate is in part attributable to a lack of chemotherapeutic agents but also other factors, first of all the infiltrative growth, the heterogeneity etc. Please correct your statement accortingly
-Further, Herrlinger at al published the NOA-09 trial on the use of Lumustine which may be included in the list of feasible chemotherapeutic agents, hence some progress has been made while I agree that the overall progress is poor
-You touch the cost-effectiveness in the conclusion of the manuscript yet fail to present data or discuss the cost-effectiveness in the discussion/results section. Please add a paragraph in the discussion referring to the costs of your analysis and potential implications on cost-effectiveness
Author Response
# Reviewer1-1:
The authors present an interesting manuscript. Some aspects should be considered before accepting the manuscript for :
-Line 51. I agree that the low survival rate is in part attributable to a lack of chemotherapeutic agents but also other factors, first of all the infiltrative growth, the heterogeneity etc. Please correct your statement accordingly
-Further, Herrlinger at al published the NOA-09 trial on the use of Lumustine which may be included in the list of feasible chemotherapeutic agents, hence some progress has been made while I agree that the overall progress is poor
##
I appreciate your kind and accurate comment, and revised the original sentence as below.
>>>
One of the reasons for this low survival rate is the lack of promising therapeutic drugs except for temozolomide (TMZ) and bevacizumab. Though some advances have been made for the limited entity of gliomas {Herrlinger, 2019 #173}, no effective new drugs have been developed against malignant gliomas for over 10 years.
# Reviewer1-2:
-You touch the cost-effectiveness in the conclusion of the manuscript yet fail to present data or discuss the cost-effectiveness in the discussion/results section. Please add a paragraph in the discussion referring to the costs of your analysis and potential implications on cost-effectiveness
##
We are also deeply grateful for your specific comment on cost-effectiveness. It is difficult to assess the cost-effectiveness of CGPT at the moment, which is by far the greatest concern for both patients and clinicians, because there are a few drugs for GBM. However, I expect many new drugs will be developed in the future. Based on your comment, we simply emphasize the clinical actionability here and changed the phrases from “time- and cost-effective” into “clinically actionable”.
>>>
we hope more clinically actionable glioma targeting CGPT be developed in the future.
Reviewer 2 Report
Omura et al have submitted as well-reasoned manuscript that flows nicely and describes the genetic landscape in a subset of patients with glioma both in Japan. The authors perform a retrospective review of glioma at their institution in which patients were recommended to have next generation sequencing (CGPT in manuscript). These patients were not representative of all patients with glioma, but were selected based on progression following standard therapy, KPS >70, or eligibility for targeted therapy based on IHC testing results (some trials required CGPT for participation). As such, the median age is much lower than that for all patients with glioma. Unsurprisingly, the authors identified a relatively high proportion of patients who were eligible for targeted therapy. They mention, appropriately, that this is enhanced by the fact that several patients underwent CGPT to confirm an actionable alteration identified on IHC (BRAF in particular).
The primary limitation of this manuscript is it's purely descriptive nature and the selected cohort of patients with recurrent glioma included in the cohort. This limits its generalizability within Japan as well as to other countries / health systems. It would be interesting to know what proportion of total patients with recurrent glioma had CGPT. It would also be interesting if the authors were able to dig deeper into the factors that prompted patients for referral to CGPT, and whether there were any in particular that were associated with a higher likelihood of identifying a targetable alteration. For example, one might think younger age could be associated with targetable alterations on NGS, but I wouldn't expect KPS to be associated.
Regardless, I recommend the authors emphasize the selected nature of these patients a little more clearly, perhaps even mention in abstract. The selection of patients for whom to send CGPT is reasonable, but the authors should make an effort to clarify that their findings don't apply to all patients with glioma, and in fact the proportion of patients with a targetable alteration is likely to be much lower given the enrichment for patients with +ICH for actionable mutations.
There are a few minor textual edits that would improve readability.
- Section 3.5 in its current form is a little confusing. I'm not sure I follow what the authors are conveying when they state "This implies that other preferable options. . . should be prioritized in these cases." Is that referring to the 4 patients with potentially targetable alterations? Also, the first sentence of the second paragraph of this section (line 241) needs to be edited for clarity.
- In lay summary, line 24, "spasmodic" is not the right word to use.
Author Response
# Reviewer 2
Omura et al have submitted as well-reasoned manuscript that flows nicely and describes the genetic landscape in a subset of patients with glioma both in Japan. The authors perform a retrospective review of glioma at their institution in which patients were recommended to have next generation sequencing (CGPT in manuscript). These patients were not representative of all patients with glioma, but were selected based on progression following standard therapy, KPS >70, or eligibility for targeted therapy based on IHC testing results (some trials required CGPT for participation). As such, the median age is much lower than that for all patients with glioma. Unsurprisingly, the authors identified a relatively high proportion of patients who were eligible for targeted therapy. They mention, appropriately, that this is enhanced by the fact that several patients underwent CGPT to confirm an actionable alteration identified on IHC (BRAF in particular).
The primary limitation of this manuscript is it's purely descriptive nature and the selected cohort of patients with recurrent glioma included in the cohort. This limits its generalizability within Japan as well as to other countries / health systems. It would be interesting to know what proportion of total patients with recurrent glioma had CGPT. It would also be interesting if the authors were able to dig deeper into the factors that prompted patients for referral to CGPT, and whether there were any in particular that were associated with a higher likelihood of identifying a targetable alteration. For example, one might think younger age could be associated with targetable alterations on NGS, but I wouldn't expect KPS to be associated.
Regardless, I recommend the authors emphasize the selected nature of these patients a little more clearly, perhaps even mention in abstract. The selection of patients for whom to send CGPT is reasonable, but the authors should make an effort to clarify that their findings don't apply to all patients with glioma, and in fact the proportion of patients with a targetable alteration is likely to be much lower given the enrichment for patients with +ICH for actionable mutations.
##
Thank you for your very important comment with highest insight on the selective nature of the patients’ cohort in this paper. As you kindly pointed out, our result may only reflect the results of the very limited patients who sufficed the criteria determined by the Japanese medical health insurance system. We also deeply understand that our result here cannot be able to be applied to all the other glioma patients, however, the patient inclusion criteria used in this paper are the ones need to be sufficed when the glioma patients consider enrolling to the clinical trials at this moment. Therefore, our result could be one of the representative remarks regarding the clinical actionability in glioma patients who are potentially eligible to clinical trials. Given this point, we added some sentences in the “Discussion” section as below, and will further appreciate if this could obtain your consent (added sentences were in red font).
>>>
Some limitations to this study are to be recognized. First, the sample of this study was small because of the selection of participants being limited to those with a good performance status. Our result may only reflect the results of the very limited patients who sufficed the criteria, and thus cannot be able to be applied to all the other glioma patients. However, the patient inclusion criteria used in this paper are the ones need to be sufficed when the glioma patients consider enrolling in the clinical trials. Therefore, our result could be one of the representative findings regarding the clinical actionability in glioma patients who are potentially eligible to clinical trials at this moment.
##
# Reviewer2-1:
There are a few minor textual edits that would improve readability.
- Section 3.5 in its current form is a little confusing. I'm not sure I follow what the authors are conveying when they state "This implies that other preferable options. . . should be prioritized in these cases." Is that referring to the 4 patients with potentially targetable alterations? Also, the first sentence of the second paragraph of this section (line 241) needs to be edited for clarity.
##
I appreciate your comment on these confusing sentences. Actually, as you kindly mentioned, these four patients had potentially targetable genomic alterations, but they needed to complete the undergoing standard therapeutic options before enrolling in the clinical trials. We have made some corrections as below to this section:
>>>
Another four (12.9%) patients could potentially have therapeutic options: this implies that they had targetable genomic mutations, but other preferable options (e.g., bevacizumab, enrollment in other more promising clinical trials) should be prioritized before enrolling to the clinical trials. Therefore, seven patients (22.6%) were proposed available clinical trials in which they could use the specific molecularly therapeutic drugs based on their genomic aberrations.
##
# Reviewer2-2:
- In lay summary, line 24, "spasmodic" is not the right word to use.
##
We appreciate your correction. We have removed “spasmodic”.
Reviewer 3 Report
This is a well-written and well-organized report on a Japanese cohort of glioma patients who underwent comprehensive genomic profiling at recurrence.
65/104 were presented at Molecular Tumor Board. 18.5 % were candidates for a therapeutic agent and 7.7 % were found to have germline mutations. Acomprehensive table is nicely presented showing the common glioma mutations (TERT, CDK2N, MTAP 50% and more; p53, EGFR, PTEN 30% and more; and NF-1, BRAF, PDGFRA, P13Kand CDK4 10%+.
Candidates needed to have tumor progression and be of KPS =/> 70%
Of 31 candidates, 7 were treated based on their genomic profile (with BRAF-inhibitor, FGFR-I, or Pembrolizumab (MSH6, high TMB).
This is interesting background data that will serve as a useful reference as we investigate the utitlity of targeted therapy for glioma in our molecular era.
1) There is no mention of response to the 7 cases treated with targeted therrapy. Was there activity? ***
(recommend including reference:
Blumenthal DT, et al. J Neurooncol. 2016 Oct;130(1):211-219. Clinical utility and treatment outcome of comprehensive genomic profiling in high grade glioma patients
2) Did their patients have (private) options for obtaining targeted therapy if there were no relevant clinical trials?
3) Would there be a rationale for considering targetedtherapy earlier in the disease course?
4) The level of English is quite good; I would suggest one change, the use of the term "spasmodic" on page 1, line 24, is incorrect. (It is not clear what you mean).
Author Response
This is a well-written and well-organized report on a Japanese cohort of glioma patients who underwent comprehensive genomic profiling at recurrence.
65/104 were presented at Molecular Tumor Board. 18.5 % were candidates for a therapeutic agent and 7.7 % were found to have germline mutations. Acomprehensive table is nicely presented showing the common glioma mutations (TERT, CDK2N, MTAP 50% and more; p53, EGFR, PTEN 30% and more; and NF-1, BRAF, PDGFRA, P13Kand CDK4 10%+.
Candidates needed to have tumor progression and be of KPS =/> 70%
Of 31 candidates, 7 were treated based on their genomic profile (with BRAF-inhibitor, FGFR-I, or Pembrolizumab (MSH6, high TMB).
This is interesting background data that will serve as a useful reference as we investigate the utitlity of targeted therapy for glioma in our molecular era.
# Reviewer3-1:
1) There is no mention of response to the 7 cases treated with targeted therapy. Was there activity? ***
(recommend including reference:
Blumenthal DT, et al. J Neurooncol. 2016 Oct;130(1):211-219. Clinical utility and treatment outcome of comprehensive genomic profiling in high grade glioma patients)
##
We deeply appreciate your recommendation of the paper by Blumenthal DT, et al., and we cited this reference in the first paragraph of the Introduction.
About activity of each molecularly targeting drugs, we apologize for not being allowed to publish the result as the clinical trials are still ongoing.
##
# Reviewer3-2:
2) Did their patients have (private) options for obtaining targeted therapy if there were no relevant clinical trials?
##
Unfortunately, there were no other options for the patients describe in our manuscript.
In Japan, mixed medical treatment with both public insurance and private options is generally not allowed. Therefore, if patients would like to use non-reimbursed drugs, they have to enroll in the appropriate clinical trials. This regulation inevitably makes some loss of opportunity for the patients who are right in the middle of the time lag between the clinical trial closing and PMDA drug approval (; PMDA is a Japanese equivalent to FDA in the U.S.). To avoid this loss of opportunity, a rescue system, named as “patient-proposed healthcare services”, is prepared, and “where there is a drug, there is a clinical trial” is de facto placed in Japan. We would further appreciate it if you kindly refer the fifth paragraph of “Discussion” section (lines 328-336).
##
# Reviewer3-3:
3) Would there be a rationale for considering targeted therapy earlier in the disease course?
##
We would like to deeply thank you again for comment on this point. We also assume this as a very important, because considering targeted therapy earlier in the disease course would make it much possible to seamlessly connect the patients from standard therapy to clinical trials.
##
# Reviewer3-4:
4) The level of English is quite good; I would suggest one change, the use of the term "spasmodic" on page 1, line 24, is incorrect. (It is not clear what you mean).
##
We appreciate your correction. We have removed “spasmodic”.
Reviewer 4 Report
Authors descripted the genomic profiling results in 104 glioma patients with commercial available tests, FoundationOne CDx Cancer Genomic Profile or OncoGuideTM NCC OncoPanel System and approved the clinical value in term of actionability and diagnostics in glioma patients. The manuscript was well-written and well-presented the data. Suggest to 1) remove the sentence of “Liquid biopsy…..extremely low” in the introduction because this is not relevant to this paper; 2) could move Table 3. Suspected germline mutations to supplementary.
Author Response
# Reviewer4-1:
descripted the genomic profiling results in 104 glioma patients with commercial available tests, FoundationOne CDx Cancer Genomic Profile or OncoGuideTM NCC OncoPanel System and approved the clinical value in term of actionability and diagnostics in glioma patients. The manuscript was well-written and well-presented the data. Suggest to
- remove the sentence of “Liquid biopsy…..extremely low” in the introduction because this is not relevant to this paper
##
We appreciate your important correction. We removed the .
##
# Reviewer4-2:
- could move Table 3. Suspected germline mutations to .
##
One of the surprising results in this paper is that there were more glioblastoma patients suspected of having a germline mutation by CGPT than expected. Both the patients and the clinicians would need this information beforehand, in order to decide whether to use genomic profiling test or not. We would like to emphasize this finding and would further appreciate if you kindly allow us to publish our comprehensive results of genomic profiling tests including the results of germline mutations.
Reviewer 5 Report
This is a well designed retrospective study focusing genomic profile in glioma patients.
However, there were numerous previous and larger studies NGS data in glioblastoma patients.
Author Response
# Reviewer5:
This is a well designed retrospective study focusing genomic profile in glioma patients.
However, there were numerous previous and larger studies NGS data in glioblastoma patients.
##
Thank you for your important comment. Though NGS data in glioblastoma patients have been numerously reported, as far as we know, limited results have been published on the real clinical actionability by using commercially available genomic profiling tests. Our manuscript is a descriptive style one, but we hope this could be one of the references on the raw results of FoudationOne Genomic Profiling Tests for glioblastoma. We would further appreciate it if you allow us to.
Round 2
Reviewer 2 Report
The authors have addressed my concerns.
Reviewer 5 Report
.